

# Spatiotemporal Recurrence Pattern of Earth's Polar Cap Variation During Geomagnetic Storms

Irewola Aaron Oludehinwa[1,2], Olasunkanmi Isaac Olusola[2], Andrei Velichko[3], RoseAnoke-Uzosike[2,4]

[1]Department of Physics, Caleb University, Lagos, Nigeria
[2]Department of Physics, University of Lagos, Lagos, Nigeria
[3]Institute of Physics and Technology, Petrozavodsk State University, 185910 Petrozavodsk, Russia.
[4]Nigerian Maritime Administration and Safety Agency (NIMASA), Victoria Island, Lagos, Nigeria.

*Correspondence to*: Irewola A. Oludehinwa (irewola.oludehinwa@calebuniversity.edu.ng)

**Abstract.** This study examined the recurrence patterns of Earth's polar cap activities in response to various geomagnetic storm intensities. Time series data of polar cap indices (PCN and PCS) for great storm, severe storm, strong storm, moderate storm, and weak storm events were analysed. Nonlinear dynamics tools, including Recurrence Plot (RP), Recurrence Rate (RR), and Length of Diagonal Line (L), were applied to the polar cap variations to identify the recurring patterns associated with the
various categories of geomagnetic storms. The RP, RR, and L effectively captured the distinct recurrence features in the PCN and PCS variations across different storm categories. During great storms, severe storms, and strong geomagnetic storms, the RPs unveils a strong deterministic structure for both PCN and PCS variations, whereas moderate and weak storms showed a rare deterministic structure of RP. Similarly, RR and L values were high during great storms, severe storms, and strong storms, however these indicators significantly decline during moderate and weak storms. These findings indicates that the recurrence
density and deterministic behaviour in the polar cap activities, intensify with increased solar wind energy input into the magnetosphere.

**Keywords:** Geomagnetic storms, Solar Wind-Magnetosphere system, Polar Cap, Recurrence Plot (RP), Recurrence Rate (RR), Length of Diagonal line (L)

**1 Introduction**

The Earth's polar cap is the region where geomagnetic field lines are open, extending into the magnetosphere and connecting to the IMF (Tulegenov et al., 2023; Zossi et al., 2019). During geomagnetic storms, driven by solar wind structures like Coronal Mass Ejections (CMEs) or Corotating Interaction Regions (CIRs), the polar cap undergoes significant variations due to enhanced solar wind-magnetosphere coupling. These variations are primarily driven by magnetic reconnection at the dayside
magnetopause, which open field lines, and subsequent reconnection in the magnetotail, which closes them (Burch & Phan, 2016; Kim et al., 2024). One of the parameters that measures the activities of the Earth's polar cap is the Northern and Southern Hemisphere Polar Cap indices (PCN and PCS).

The polar cap indices endorsed by the International Association of Geomagnetism and Aeronomy (IAGA) represent the coupling of solar wind energy and momentum injected into the magnetosphere and is used to monitor the variations of the
Earth polar cap activities (Stauning, 2012, 2018). These polar cap activities are generated from geomagnetic recordings made





at Thule (Quanaaq) in Greenland as Polar Cap Northern (PCN) Hemisphere and Vostok in Antarctica as Polar Cap Southern (PCS) Hemisphere.

Notably, the variation of these polar cap activity produced by the solar wind impact on the magnetosphere possesses some coherent structures, which are complex multiscale and chaotic. They tend to expand during geomagnetic storms due to
enhanced solar wind-magnetosphere interactions. Despite the fluctuating nature of the polar cap variations. These variations exhibit some repetitive patterns in response to geomagnetic storms that needs to be investigated. Therefore, complex system methods such as Recurrence Plot (RPs) and Recurrence Quantification Analysis (RQA) characterizing nonlinear processes in a dynamical system are capable to capture the recurring patterns in the polar cap variations (Bonizzi et al., 2019; Marwan & Kraemer, 2023). The information obtained from the recurrence measures regarding the repetitive patterns in the Earth polar
cap variation will reveal the nonlinear, multiscale nature of magnetosphere-Ionosphere dynamics, aiding model development for Geospace extreme events. This idea forms the bedrock of this study to examine the recurrence pattern (repetitive behaviour) in the polar cap fluctuations in response to Heliospheric disturbances.

Several studies have been conducted on the Hemispheric differences of the Earth polar cap. For example, Troshichev and Sormakov, (2018) studied the polar cap magnetic activity (PCN and PCS) indices as a proxy of the solar wind energy that
entered into the magnetosphere. The study considered that polar cap variation usually follows the changes of the Interplanetary Electric Field (EKL). They applied correlation analysis between PC index and the estimated EKL to examine the course of magnetic disturbances. The analysis showed that magnetic activity in the winter polar cap (PC winter) reveals statistically more correct results than that in the summer polar cap (PC summer). Furthermore, correlation between the PC winter index and the estimated EKL unveils a good correlation value (R>0.5) in about 80% of the substorms events investigated. Troshichev,
(2022) examined the relationship of PC index to the solar wind electric field (EKL). PC index data of solar cycle 23 and 24 were analysed in relation to various solar wind parameters. The author considers the relationship of PC index to the magnetospheric field-aligned current (FAC) and to solar wind dynamic pressure. It was reported that the polar cap magnetic activity is controlled by the solar wind electric field (EKL) affecting the magnetosphere. In addition, the author reported that the PC index behaviour certifies development and intensity of the magnetospheric disturbances. Lockwood, (2023) studied the
extent of the differences between the Northern and Southern Hemispheric Polar Cap indices, (PCN and PCS). The author used PCN and PCS indices from 1998-2018 to show the potential effects of the slightly different and changing, magnetic coordinates of the two magnetic stations employed, Thule (Quanaaq) in Greenland as PCN and Vostok in Antarctia as PCS. It was shown that the agreement in overall behaviour of the two indices is very close indeed. However, PCS consistently correlates slight better with solar wind parameters than PCN. Park et al. (2024) investigate the statistical features of PCN and PCS in response
to various interplanetary conditions. The concurrent PCN-PCS pairs from 1998-2002 and 2024-2018 were divided based on their sign type (positive-positive, negative-negative, negative-positive and positive-negative PCN-PCS pairs). Their findings revealed that PCN-PCS pair data provide local views about the solar wind-magnetosphere-ionosphere (SW-M-I) coupling system with different control efficiencies of IMF orientation and season depending on the middle latitude (MLT) location of the stations. Also, the works on the recurrence analysis on geomagnetic activities during geomagnetic storms were reported.
Oludehinwa et al. (2018) used the recurrence analysis to examined the nonlinearity effect in the disturbed storm time (Dst) during various categories of geomagnetic storm. The authors reported that the Dst signals behave in stochastic manner during minor geomagnetic storms while a deterministic manner are exhibited during increasing magnetic stormy periods. Donner et al. (2018) also applied the recurrence analysis to unveil the nonlinear features of the hourly Dst index during the period with magnetic storms and normal variability. They found that recurrence quantification analysis (RQA) distinguishes between
storms and non-storms times even better than other considered nonlinear characteristics like symbolic dynamics-based entropy. In 2019, Donner and co used RQA to obtain the complementary measures that serves as markers of different physical processes underlying quiet and storm time magnetospheric dynamics. It was shown that the RQA approach discriminates the magnetospheric activity in response to solar wind forcing.

However, the investigation of repetitive patterns in polar cap variations in response to varying intensities of geomagnetic
storms remains an open research area. Also, the idea of using the concept of nonlinear dynamics to examine the repetitive patterns in the polar cap indices (PCN and PCS) have not been considered in the literature to the best of our knowledge. Therefore, the study seeks to examine the recurring patterns in the polar cap north (PCN) and south (PCS) in response to




different intensities of geomagnetic storms. Our study implements the concept of nonlinear dynamics namely recurrence analysis, where the times at which the recurrence occurs in a time series data is accounted. Recurrence Plot (RPs) reveals patterns of how a complex system such as solar wind-magnetosphere interactions revisits similar states over time while Recurrence Quantification Analysis (RQA) is a set of quantitative measures derived from RPs to characterize the underlying dynamics of the system (Donner et al., 2019; Marwan et al., 2007; Oludehinwa et al., 2018; Oludehinwa et al., 2021). RPs and RQA have been proven to be useful diagnostic tools in many areas of discipline (Bonizzi et al., 2019; Dimitriev et al., 2020). We describe the structure of this paper such that our methods of data acquisition are described in section 2. Also, the nonlinear analysis that we employed in this investigation are explained in details. In section 3, we unveiled our results and observation in section 4, the discussion and conclusion in section 5.

**2.0 Data Acquisition and the Concept of Nonlinear Dynamics**

The polar cap indices (PCN, and PCS) used in this study was obtained from the archive of the International Service of Geomagnetic Indices (ISGI), https://isgi.unistra.fr. To carry out this study, we selected different categories of geomagnetic storm events based on their storm intensities and the corresponding PCN and PCS indices dataset of the events was acquired to investigate the recurring pattern in the Earth polar cap during storm-driven variability. In determining the actual intensities of the selected categories of geomagnetic storm events used in this study, we considered $SYMH$ ($D_{st}$ in minutes) accessed from Space Physics Data facility, NASA Goddard Space Flight Centre (https://omniweb.gsfc.nasa.gov/form/omni_min.html) because the polar cap indices are in 1-minutes resolution.

The geomagnetic storm events selected were categorized in accordance to the work of (Borovsky & Shprits, 2017; Loewe, 1997; Oludehinwa et al., 2021). Such that minimum $SYMH$ ranging with $SYMH < -350nT$ was categorized as great storms, while $-200nT > SYMH > -350nT$ was classified as severe storms. Storms with minimum $SYMH$ of $-100nT > SYMH > -200nT$ was referred to as strong storms and minimum $SYMH$ within $-50nT > SYMH > -100nT$ are classified as moderate geomagnetic storms. Minimum $SYMH$ within $-30nT > SYMH > -50nT$ are categorized as weak geomagnetic storms as shown in Table 1.

Table 1: The categories of geomagnetic storms with dates of occurrence investigated in this study

| S/No | Categories of Geomagnetic Storm Events investigated | Date of Storm Occurrence | Intensities of the Geomagnetic storms ($SYMH$) | |
|------|------|------|------|------|
| 1 | Great storms | May 10th, 2024 | Storm event 1 | -354nT |
| | | October 10th, 2024 | Storm event 2 | -390nT |
| 2 | Severe storms | March 17th, 2015 | Storm event 1 | -234nT |
| | | December 15th, 2006 | Storm event 2 | -210nT |
| 3 | Strong storms | March 24th, 2023 | Storm event 1 | -162nT |
| | | September 12th, 2024 | Storm event 2 | -130nT |
| 4 | Moderate storms | April 27th, 2023 | Storm event 1 | -56nT |
| | | September 25th, 2024 | Storm event 2 | -69nT |
| 5 | Weak storms | January 6th, 2008 | Storm event 1 | -37nT |
| | | February 22nd, 2021 | Storm event 2 | -35nT |



**2.1 Recurrence Plot (RPs):** Is a graphical tool used to visualize and analysed the recurrence of states in a dynamical system. This concept was introduced by Eckmann et al. (1987) to study the behaviour of nonlinear dynamical systems revealing
patterns such as periodicity, chaos, or trends. The RPs is based on recurrence matrix where each element $(i, l)$ indicates whether the state of a system at time $i$ is close to the states at time $j$ within a defined threshold $(\varepsilon)$. If the states are close $R(i, j) = 1$, a black dot is pointed on the RP, otherwise the RP is left blank. Therefore, our study implements this concept to capture and reveal the recurrence density of the polar cap variation in response to different intensities of geomagnetic storms in visualize format. The mathematical expression of RPs is described below:

Given a PCN and PCS time series as

$$x_i = (x_1, x_2, \ldots, x_n), \quad i = 1, 2, \ldots, 1440 \qquad\qquad 1$$

Embedding the PCN and PCS time series into higher-dimensional phase-space using time-delay embedding to capture the polar cap dynamics.

$$\bar{x}_i = \left(x_i, x_{i+\tau}, \ldots, x_{i+\tau(m-1)}\right) \qquad\qquad 2$$

Where $m$ is the embedding dimension and $\tau$ is the time delay.

Then, the distance between all pairs of points in the phase space is computed. This is done by applying a threshold $(\varepsilon)$ to determine recurrence: if the distance between points $x_i$ and $x_j$ is less that $\varepsilon$, marks $R(i, j) = 1$, otherwise $R(i, j) = 0$.

$$R_{i,j} = \Theta\left(\varepsilon_i - \left\|\bar{x}_i - \bar{x}_j\right\|\right) \quad i, j = 1, 2, \ldots, N \qquad\qquad 3$$

Where $N$ is the total numbers of considered states, $\bar{x}_i, \bar{x}_j$ are considered states, $\varepsilon_i$ is the threshold distance, $\|.\|$ is the norm,
and $\Theta$ is the Heaviside function. Our recurrence analysis was computed using the embedding dimension $(m = 15)$ and time delay $(\tau = 6)$ to reveal the polar cap recurrence pattern during different categories of geomagnetic storms shown in Table 1.

The Recurrence Quantification Analysis (RQA) is deduced to measure the structure and patterns in the RPs. RQA presents the number and duration of recurrences to characterize the polar cap behaviour. The RQA comprises of several quantities measures





such as Recurrence Rate (RR), Laminarity (LAM), Entropy (ENTR), Determinism (DET), Average Diagonal Line Length (L),
Trapping Time (TT). However, our study of polar cap behaviour was restricted to RR and L.

**2.2 The Recurrence Rate (RR)**:  measures the density of recurrence points in the RPs indicating how often states recurs. It is
mathematically express as;

$$RR = \frac{1}{N^2} \sum_{i,j=1}^{N} R(i,j) \qquad\qquad 4$$

Where $R(i,j)$ is the recurrence matrix and $N$ is the number of points.

**2.3 The Average line length of the diagonal (L):** quantifies the average length of diagonal lines in a RP. It represents the
average length of diagonal lines in the RP that are longer than or equal to a separated minimum length. The Average line length
of the diagonal (L) is mathematically expressed as:

$$L = \frac{\sum_{l=l_{min}}^{N} l.P(i)}{\sum_{l=l_{min}}^{N} P(i)} \qquad\qquad 5$$

Where $l$ is the length of a diagonal line and $P(l)$ is the number of diagonal lines of length ($l$) in the RP. $l_{min}$ represent the
minimum line length considered while $N$ is the maximum possible line length in the RP.

**3.0 Results and Observations**

We show in Figure 1(a-e), the variation of the polar cap indices Polar Cap North (PCN) and Polar Cap South (PCS) during
different categories of geomagnetic storms. Panel 1(a) present the PCN and PCS variation during great storm, while panel 1(b)
unveil the PCN and PCS variation during severe storm. The panel 1(c) display the variation of the PCN and PCS at strong
storm and in panel 1(d), we depict the PCN and PCS variation during moderate storm. The PCN and PCS variation during
weak storm is shown in panel 1(e). At the different categories of geomagnetic storms depicted by the polar cap variation, it
was seen that the polar cap signals exhibit some coherent structure in its underlying dynamics. The information of these inherit
coherent structures is revealed through recurrence analysis.

Figure 2(a-d) presents the Recurrence Plots (RPs) for PCN and PCS indices during great storms on May 10, 2024, and October
10, 2024. Specifically, Figure 2(a) shows the recurrence density in the RP for PCN fluctuations during the May 10, 2024,
storm, while Figure 2(b) displays the RP for PCS fluctuations during the same event. Similarly, Figure 2(c) illustrates the
recurrence density for PCN during the October 10, 2024, storm, and Figure 2(d) depicts the RP for PCS during the October
10, 2024, storm. The RPs observation for both PCN and PCS during these great storms reveals a broad distribution of
recurrence points, indicating an elongated recurring pattern in the solar wind-magnetosphere system's dynamics as storm
intensity increases. This suggests a heightened degree of similarity in the behaviour of the solar wind-magnetosphere coupling
process during great storms. Notably, the recurrence point density in RPs for PCS is higher than that for PCN, indicating
stronger recurrence patterns in the southern hemisphere.

Figure 3(a-d) shows the RPs for PCN and PCS indices during severe geomagnetic storms on March 17, 2015, and December
15, 2006. For instance, Figure 3(a) shows the RP for PCN during the March 17, 2015, storm, while Figure 3(b) displays the





RP for PCS during the March 17, 2015, storm. Figure 3(c) illustrates the RP for PCN during the December 15, 2006, storm, and Figure 3(d) depicts the RP for PCS during the December 15, 2006. The RPs observation for these severe geomagnetic storms captured a wide distribution of recurrence points, with a high density of recurrence points observed in all the severe storms investigated. These features indicate a strong degree of similarity in the dynamics of polar cap activities as stormy

variability intensifies.

Figure 4(a-d) displays the RPs for PCN and PCS indices during strong geomagnetic storms on March 24, 2023, and September 12, 2024. The Figure in panel 4(a) depicts the RP for PCN during March 24th, 2023, and Figure 3(b) is the RP observation of PCS variation during March 24th, 2023. Figure 4(c) shows the RP for PCN during the September 12, 2024, storm, while Figure 4(b) presents the RP for PCS during the September 12, 2024. The observed RPs for these strong geomagnetic storms reveal a

high density of recurrence points with a wide distribution. This broad spread of recurrence points in the RPs indicates consistent similarity dynamics in the polar cap activities during strong geomagnetic storms.

Figure 5(a-d) presents the RPs for PCN and PCS indices during moderate geomagnetic storms on April 27, 2023, and September 25, 2024. Figure 5(a) shows the RP for PCN during the April 27, 2023, storm, while Figure 5(b) displays the RP for PCS during the same event. Figure 5(c) illustrates the RP for PCN during the September 25, 2024, storm, and Figure 5(d)

depicts the RP for PCS during the September 25, 2024, storm. The RPs for these moderate geomagnetic storms reveal a sparse distribution of recurrence points, with fewer recurrence points observed compared to great, severe and strong storms. This sparse distribution indicates reduced similarity dynamics in the polar cap activities as stormy intensity declines.

Figure 6(a-d) shows the RPs for PCN and PCS indices during weak geomagnetic storms on January 6, 2008, and February 22, 2021. Distinctly, Figure 6(a) depicts the RP for PCN during the January 6, 2008, storm, while Figure 6(b) displays the RP for

PCS during the January 6, 2008. Figure 6(c) show the RP for PCN during the February 22, 2021, storm, and Figure 6(d) depicts the RP for PCS during the February 22, 2021. The RPs observation for these weak geomagnetic storms captured an extremely low density of recurrence points, indicating a rare deterministic behavior in the polar cap activities. This sparse recurrence pattern reflects significantly weakened similarity dynamics in polar cap variations, corresponding to the decrease in stormy intensity.

**3.1 Length of Diagonal Line (L) observations for PCN and PCS Variations during Super-Intense, Intense, Major, Moderate, and Weak Geomagnetic Storms**

Figure 7 presents a bar chart illustrating the Length of Diagonal Line (L) in RPs for PCN and PCS variations across different geomagnetic storm intensities. The blue bars represent L values for PCN during storm event 1, magenta bars illustrate L values for PCS during storm event 1, yellow bars indicate L values for PCN during storm event 2, and the grey bars depict L values

for PCS during storm event 2. The highest L values are observed during great geomagnetic storms, with the most prominent value seen in PCS for storm event 1, followed by PCN for storm event 2. During severe geomagnetic storms, PCS exhibits the higher L values. It was noted that lower values of L were obtained during severe storms when compared to the values obtained during great storms. Furthermore, during strong geomagnetic storms, L values for both PCN and PCS decline further compared to great and severe storms. A further decrease in L were also noticed during moderate geomagnetic storms, while the lowest



L values were observed during weak geomagnetic storms. We remark that the high L values during great, severe, and strong geomagnetic storms are indications of prolonged similarity dynamics in the solar wind-magnetosphere-ionosphere coupling, reflecting stronger deterministic behavior as stormy intensity increases. On the other hand, the progressive decline in L values during moderate and weak storms suggests shorter similarity dynamics, indicating less deterministic behavior as geomagnetic storm activity diminishes.

**3.2 Recurrence Rate (RR) Observations for PCN and PCS Variations during Super-Intense, Intense, Major, Moderate, and Weak Geomagnetic Storms**

Figure 8 presents a bar chart illustrating the RR observations for PCN and PCS indices during various geomagnetic storm intensities. The blue bars represent RR values for PCN during storm event 1, while magenta bars indicate RR values for PCS during the same event. Yellow bars depict RR values for PCN during storm event 2, and grey bars show RR values for PCS

during storm event 2. During great geomagnetic storms, both PCN and PCS reveals high RR values, with PCS showing notably higher values, indicating a greater recurrence density in the Southern Hemisphere's polar cap compared to the Northern Hemisphere. During severe geomagnetic storms, we observe that RR values remain high but are lower than those obtained during great storms. At strong geomagnetic storms, RR values for both PCN and PCS were observed to decrease compared to great and severe storms. Furthermore, a further decline in RR values during moderate geomagnetic storms was noticed as seen

in Figure 7. Finally, the lowest RR values are observed during weak geomagnetic storms, reflecting minimal recurrence density in the polar cap activities as the solar wind energies injecting into the magnetosphere declines.

**4.0 Discussion and Conclusion**

The RP results of the polar cap activities during geomagnetic storms reveals distinct patterns tied to storm intensity. Great

storms, severe storms, and strong storms reveals a robust deterministic structure in RPs, reflecting heightened deterministic behaviour in the polar cap dynamics as solar wind energy injection into the magnetosphere intensifies. This is evident from the enhanced stormy variability. On the other hand, RP observations of PCN and PCS variation during moderate and weak geomagnetic storms show a sparse distribution of recurrence points, indicating a less or diminishing deterministic behaviour as the solar wind energy injection into magnetosphere reduces. The RPs observation in the polar cap variation during these

different categories of geomagnetic storms is in agreement with Oludehinwa et al. (2018) that reported that as the geomagnetic storm increases, the RP depicts more deterministic behaviour while as the geomagnetic storm intensity declines, the magnetospheric dynamics tends to exhibit stochastic behaviour. This observed less deterministic structure of RP in the polar cap variation are synonymous with reduced solar wind energy penetration, likely due to an outward shift of the magnetopause under low solar wind dynamic pressure, which decreases the magnetosphere's exposure to solar wind (Burch & Phan, 2016;

Goldstein et al., 2005).

Further results from the recurrence quantification analysis reveals that RR values for PCN and PCS increases with storm intensity. For instance, the highest RR values was observed during great storms, particularly for PCS variations. Severe and strong storms show progressively lower RR values compared to great storms, indicating a gradual increase in similarity



dynamics as solar wind energy input strengthens. However, RR values decline significantly during moderate and weak storms,
reflecting diminished similarity in polar cap dynamics. The length of diagonal lines (L) in the RPs observation reveals similar
RR trends. Great storms yield the longest L values, with slight reductions observed in severe and strong storms. Moderate and
weak storms exhibit short L values, revealing increased chaotic behaviour during moderate and weak storms. Notably, the
shorter, the length of diagonal lines in RPs, the more chaotic behaviour in a dynamical system (Marwan et al., 2007; Marwan
& Kraemer, 2023). This observation of L values in the RPs during different categories of suggests that the polar cap activities
are highly chaotic during moderate and weak storms compared to strong, severe and great storms

## Acknowledgements

The authors would like to appreciate the International Service of Geomagnetic Indices (ISGI) for making available the polar
cap indices (PCN and PCS) data available for research purpose. We also extend our appreciation to Space Physics Data facility,
NASA Goddard Space Flight Centre for making available the access to geomagnetic storm indices namely SYM-H.

## Data Availability

The polar cap indices (PCN and PCS) data used in this study are publicly available and provided by the International Service
of Geomagnetic Indices (ISGI), website  https://isgi.unistra.fr.

## Code availability

The MATLAB code used to perform the recurrence analysis can be access at Potsdam Institute for Climate Impact Research
(PIK) library, https:// https://tocsy.pik-potsdam.de/crp.php (accessed on 18th May, 2025)

## Author contributions

**IAO** developed the idea behind the problem being solved, supervise the project, analyzed the data; developed the codes;
interprets and discuss the results and draft the manuscript; **OOI** supervise the project and contribute to the drafting of the
manuscript; **AV** discuss, read and made useful comments to the manuscript; **RA** read and made useful comments to the
manuscript.

## Competing interests

The authors declare they have no conflict of interest.




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

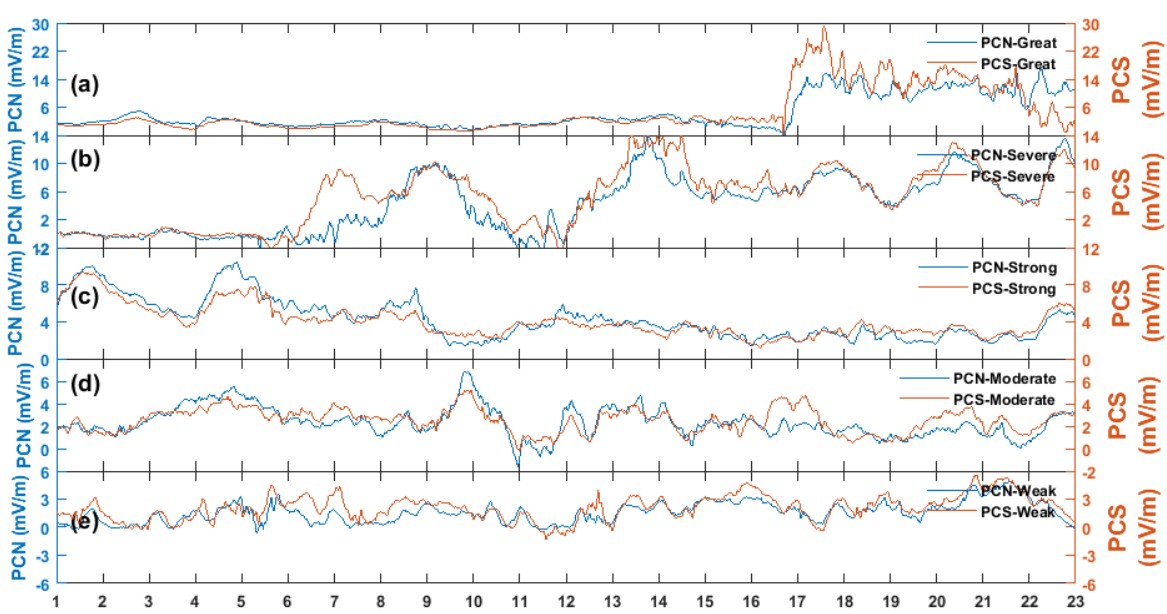


Figure 1: A sample of polar cap variation: (a) PCN (in blue color) and PCS (in magenta color) during great storm of May 10, 2024 (b) PCN (in blue color) and PCS (in magenta color) during severe storm of March 17, 2015 (c) PCN (in blue color) and PCS (in magenta color) during strong geomagnetic storm of March 24, 2023 (d) PCN (in blue color) and PCS (in magenta color) during moderate geomagnetic storm of April 27, 2023 (e) PCN (in blue color) and PCS (in magenta color) during minor

geomagnetic storm of January 6, 2008





Figure 2: The Recurrence Plots (RPs) for PCN and PCS during great storms of May 10th, 2024 and October 10th, 2024: (a) The RP for PCN on May 10th, 2024 (b) The RP for PCS on May 10th, 2024 (c) The RP for PCN variation on October 10th, 2024 (d) The RP for PCS variation on October 10th, 2024.



Figure 3: The Recurrence Plots (RPs) for PCN and PCS during severe storms of March 17th, 2015 and December 15th, 2006: (a) The RP for PCN on March 17th, 2015 (b) The RP for PCS on March 17th, 2015 (c) The RP for PCN variation on December 15th, 2006 (d) The RP for PCS variation on December 15th, 2006.







Figure 4: The Recurrence Plots (RPs) for PCN and PCS during strong storms of March 24th, 2023 and September 12th, 2024: (a) The RP for PCN on March 24th, 2023 (b) The RP for PCS on March 24th, 2023 (c) The RP for PCN variation on September 12th, 2024 (d) The RP for PCS variation on September 12th, 2024.






Figure 5: The Recurrence Plots (RPs) for PCN and PCS during moderate geomagnetic storms of April 27th, 2023 and September 25th, 2024: (a) The RP for PCN on April 27th, 2023 (b) The RP for PCS on April 27th, 2023 (c) The RP for PCN variation on September 25th, 2024 (d) The RP for PCS variation on September 25th, 2024.






Figure 6: The Recurrence Plots (RPs) for PCN and PCS during weak geomagnetic storms of January 6th, 2008 and February 22nd, 2021: (a) The RP for PCN on January 6th, 2008 (b) The RP for PCS on January 6th, 2008 (c) The RP for PCN variation on February 22nd, 2021 (d) The RP for PCS variation on February 22nd, 2021.





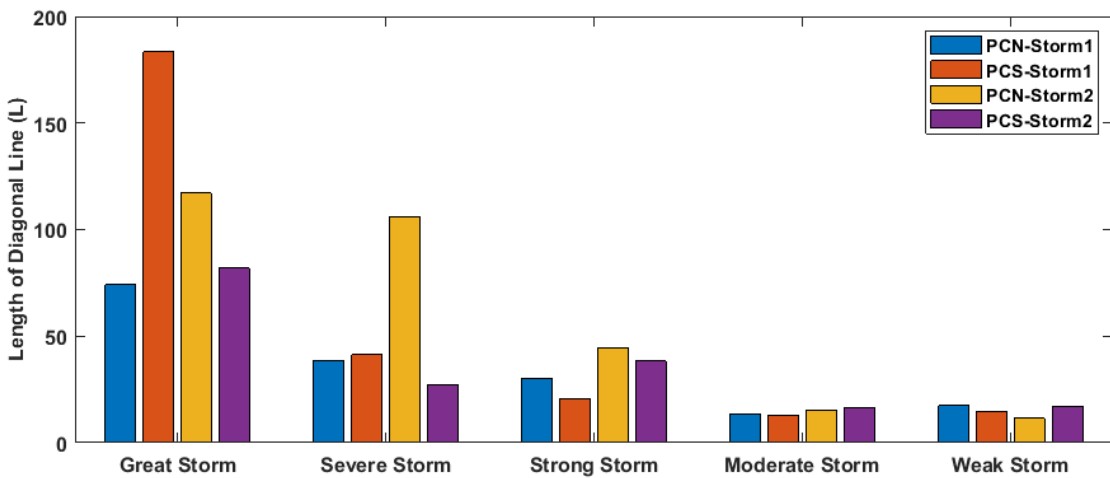


Figure 7: The Bar Chart of Length of Diagonal Line (L) for PCN and PCS at great storm, severe storm, strong storm, moderate storm and weak geomagnetic storm.

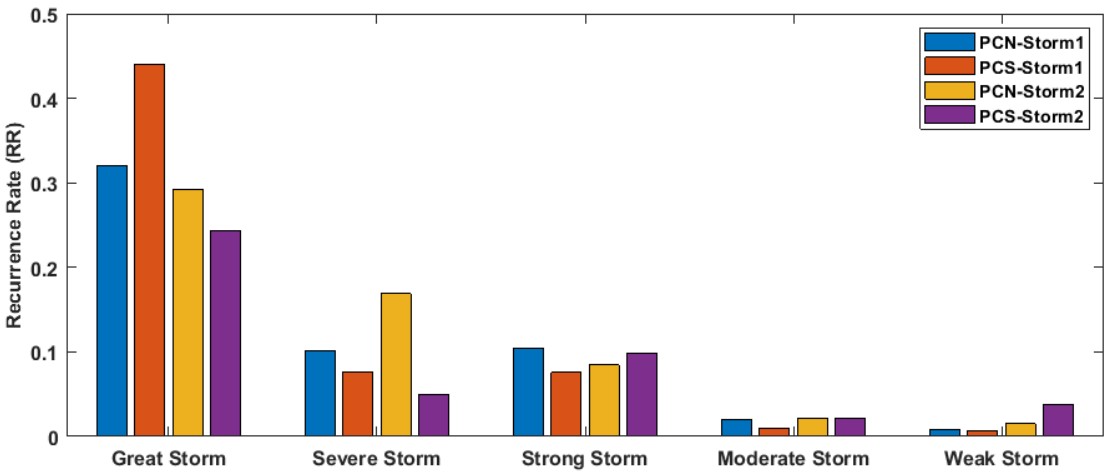

Figure 8: The Bar Chart of Recurrence Rate (RR) for PCN and PCS at great storm, severe storm, strong storm, moderate storm and weak geomagnetic storm.
