# Peer review of "Spatiotemporal Recurrence Pattern of Earth's Polar Cap Variation During Geomagnetic Storms"

_EGUsphere, 2025_

## Author Comment (AC1)

**Dear Reviewer**,

Thanks for your thorough review of our manuscript. Your contribution has enhanced the improvement of the manuscript. Below is our point-by-point response to your comments.

**Comment 1**: It is unclear if there are any reasons that the authors examine ONLY TWO geomagnetic storm events for each of the five categories.

**Response to Comment 1**: In the revised manuscript, we stated the reason why we select only two geomagnetic storm events for each of the five categories in manuscript line 105-110 as: We selected two geomagnetic storm events for each of the categories of geomagnetic storm because the time series of the polar cap index corresponding to each storm event comprises of both PCN and PCS. As a result, we focus our investigation to two storm events for each categories of geomagnetic storms.

**Comment 2:** This reviewer recommends providing the start time and end time of the geomagnetic storm events (rather than the storm occurrence date) and the time when the SYM-H index is minimum in Table 1.

**Response to Comment 2**: In the revised manuscript, we have included in Table 1, the time when the SYMH-H index is minimum.

**Comment 3**: Equation 1 ($x_i = (x_1, x_2, \ldots, x_n)$, $i = 1, 2, \ldots, 1440$) #1: Because this study focuses on the recurrence pattern of the variations in the (1-min resolution) polar cap indices "during geomagnetic storms", the numbers of the time series data ($i$) in both the PCN and PCS indices for geomagnetic storm events can vary depending on the number of minutes in the time interval of the events. Therefore, all analyses used in this study could be performed using the polar cap indices during the geomagnetic storm intervals, instead of the storm occurrence dates. This reviewer cannot find any reasons that the numbers of data points in both polar cap indices are fixed to 1440 (equivalent to 1-day data).

**Response to comment 3**: In the revised manuscript line 115-120 we have included the reason why the numbers of data points in both polar cap indices (PCN and PCS) are fixed to 1440 (equivalent to 1-day data) as: Our motive is to capture the entire repetitive patterns in the polar cap signal corresponding to the day of the storm event. Each storm event comprises of three phases namely the initial, main and recovery phases. This three phases make-up the entire dynamics of the storm event. Partitioning PCN and PCS index time series of each storm events into the time interval of initial, main and recovery phases make the study difficult, because the varying time interval of each phases differs, thereby making the data point too short for computation of recurrence analysis. This observation prompts our study to focus on the 1-minute resolution of the polar cap index. 1-minute resolution time series of PCS and PCN index corresponding to great, severe, strong, moderate and weak geomagnetic storms were acquired. Each day of the event in 1-minute resolution is equivalent to 1440 data points. This large data points in minute resolution is

computationally advantageous for the RP and RQA to capture the dynamical information of the polar cap index.

**Comment 4:** It is also worth noting that the X-axis range in Figure 1 (01:00-23:00 UT in this version, rather than 00:00-24:00 UT) as well as the X-axis range and Y-axis range in Figures 2-6 can vary according to the time interval of the events.

**Response to Comment 4**: In the revised manuscript. We have made the correction on X-axis range to 00:00-24:00 UT in Figure 1. Similar to our response in comment 3, our focus is to capture the entire dynamics of the day of the event, not the time interval of the events. This is the primary reason our study focuses on 1-minute resolution time series.

**Comment 5**: Equation 1 #2: It is difficult to understand what "xi = (x1, x2, …, xn)" means. Equation 1 needs to be rewritten more clearly.

**Response to Comment 5**: In the revised manuscript line 130, we have re-written the equation as: Suppose we assume that the sequence of a given PCN and PCS index is represented as $N$ data points.

$$[x_i, i = 1, 2, … N]$$

**Comment 6**: Line 111: It is unclear which value is used as the threshold distance ($\varepsilon$).

**Response to Comment 6**: We have included in the revised manuscript line 140 as: Our recurrence analysis was computed using the threshold distance ($\varepsilon = 1.5$) while the embedding dimension ($m = 15$) and time delay ($\tau = 6$) to reveal the polar cap recurrence pattern during different categories of geomagnetic storms shown in Table 1.

**Comment 7**: Lines 125-126: How do the authors determine 15 as the embedding dimension (m = 15) and 6 as the time delay ($\tau = 6$)?

**Response to Comment 7**: In the revised manuscript line 140-145, we have explained how the embedding dimension and time delay were determined as: The false nearest neighbor and average mutual information shown in Figure 2 and 3 were used to determine the value of embedding dimension ($m = 15$) and time delay ($\tau = 6$) used in the computation of RP and RQA. The optimal embedding dimension is determined when the varying embedding dimension against false nearest neighbor maintains a steady values while the time delay is determined at the first local minimum curve of the average mutual information.

**Comment 8:** Section 3 #1: The results shown in Figures 2-6 can be provided in terms of visible elements such as diagonal lines, vertical/horizontal lines, clustering of points, and so on.

**Response to comment 8**:  We have correction to the results in the revised manuscript to describe our results in terms of visible element of the RPs. For instance, in manuscript line 170-180 as: "The RPs observation for both PCN and PCS during these great storms reveals a widest cluster of

recurrence points, indicating an elongated recurring pattern in the solar wind-magnetosphere system's dynamics as storm intensity increases." And in the entire revised manuscript.

**Comment 9:** Section 3 #2: It is needed to elaborate on the locations in Figures 2-6 that the authors pointed out in the text (for example, "… it was seen that the polar cap signals exhibit some coherent structure in its underlying dynamics." on Lines 147-148,

**Response to Comment 9**: We have included in the revised manuscript line 170-174, the elaboration on ""… it was seen that the polar cap signals exhibit some coherent structure in its underlying dynamics." as: At the different categories of geomagnetic storms corresponding to the polar cap index time series, it was observed that the polar cap signals exhibit some coherent structures, i.e. depiction of self-organized patterns and complex fluctuations signatures in the signals of the polar cap index was evident.  The dynamical information of these complex fluctuation signatures in the signal of the polar cap index is revealed through RP and RQA.

**Comment 10:** "The RPs observation for these severe geomagnetic storms captured a wide distribution of recurrence points, with a high density of recurrence points observed in all the severe storms investigated." on Lines 162-164, and so on).

**Response to Comment 10:** The comment "a wide distribution of recurrence points, with a high density of recurrent points observed in all the severe storms investigated" has been corrected in the revised manuscript line 187-190 as:

"The RPs observation for these severe geomagnetic storms captured a wider cluster of recurrence points in the RP for PCN and PCS index in all the severe storms investigated. These features of a wide cluster of recurrence points during severe storm indicates a strong degree of similarity in the dynamics of polar cap activities as stormy variability intensifies."

**Comment 11**:  Section 4 (Discussion and Conclusion): Only the discussion of the results obtained from the recurrence plots and from the recurrence quantification analysis (in terms of RR and L) has been provided in Section 4. It is also needed to provide the conclusion of this study in this section.

**Response to Comment 11**: In the revised manuscript 280-295 we have included the Conclusion in a separate section as: This study had examined the recurrence patterns in the signals of the Earth polar cap index corresponding to great, severe, strong, moderate, and weak geomagnetic storms using recurrence plot (RP) and recurrence quantification analysis (RQA) techniques. The recurrence rate (RR) and length of diagonal line (L) was able to capture the distinct recurrence features in the PCN and PCS signals during the various categories of geomagnetic storms. The RP captures a widest, wider, and wide clusters of recurrence points in the RP of PCN and PCS corresponding to great, severe and strong geomagnetic storms while a spare and low clusters of recurrence points in the RP of PCN and PCS was observed during moderate and weak geomagnetic storms. These observation of widest, wider, and wide clusters of recurrence points in the RP of PCN and PCS corresponding to great, severe and strong geomagnetic storms signifies that the dynamics of the Earth polar cap possesses a strong deterministic structure as the solar wind energy injection into the magnetosphere intensifies. The observed low and lower clusters of recurrence

points in the RP of PCN and PCS signifies that the dynamics of the Earth polar cap possesses a rare deterministic structure as the influx of solar wind energy into the magnetosphere declines. Furthermore, the observed RR and L depict highest, higher, and high values during great, severe, and strong geomagnetic storms while a low and lower values of RR and L were observed at moderate and weak geomagnetic storms.

**Comment 12**: Title: The abstract and main body of this manuscript lead to a question about why the authors put "Spatiotemporal" in the title.

**Response to Comment 12**: In the revised manuscript, we have removed the comment "Spatiotemporal" from the title of the manuscript.

**Comment 13:** Line 28: In "… the polar cap undergoes significant variations …", please elaborate on the kind of variations the polar cap undergoes significantly.

**Response to Comment 13**: The above comment has been elaborated in the revised manuscript line 28 as: "During geomagnetic storms, driven by solar wind structures like Coronal Mass Ejections (CMEs) or Corotating Interaction Regions (CIRs), the polar cap index recordings exhibit significant complex fluctuation signatures in its underlying dynamics due to enhanced solar wind-magnetosphere coupling."

**Comment 14**: Lines 31-32: In "One of the parameters that measures the activities of the Earth's polar cap is the Northern and Southern Hemisphere Polar Cap indices (PCN and PCS).", please elaborate on which activities of the Earth's polar cap.

**Response to Comment 14**: We have elaborate on the activities of the Earth's polar cap in the revised manuscript line 29-35 as: One of the measures that monitor the activities of the Earth's polar cap such as magnetospheric convection, large-scale circulation pattern from the dayside magnetopause across the polar cap and into the nightside magnetotail, ion outflow, magnetic reconnection, and precipitation of electrons accelerated by localized electric fields is the Northern and Southern Hemisphere Polar Cap indices (PCN and PCS).

**Comment 15:** Lines 38-39: Detailed description about "some coherent structures, which are complex multiscale and chaotic" is needed. Moreover, it is unclear what "They" indicates.

**Response to Comment 15**: The comment "some coherent structures, which are complex multiscale and chaotic" has been well-explained in the revised manuscript line 42-45 as: The rapid changes in the polar cap **signals** driven by the solar wind impact on the magnetosphere possesses some coherent structures, which are complex multiscale and chaotic. Coherent structures refer to the self-organized patterns that emerge due to the nonlinear interaction between the solar wind, ionosphere and magnetosphere. Notably, the rapid changes in the polar cap are depicted in the signal**s** of polar cap index in the form of complex fluctuating signatures. These fluctuating signatures exhibit some repetitive patterns in response to different categories of geomagnetic storms that needs to be investigated.

**Comment 16**: Line 65: "… and 2024-2018 …" → "… and 2004-2018 …"

**Response to Comment 16**: The above statement has been corrected in the revised manuscript line 67-68 as: The concurrent PCN-PCS pairs from 1998-2002 and 2004-2018 were divided based on their sign type (positive-positive, negative-negative, negative-positive and positive-negative PCN-PCS pairs).

**Comment 17**: Line 68: "… middle latitude (MLT) …" → "… magnetic local time (MLT) …"

**Response to Comment 17**: The above statement has been corrected in the revised manuscript line 70-72 as: Their findings revealed that PCN-PCS pair data provide local views about the solar wind-magnetosphere-ionosphere (SW-M-I) coupling system with different control efficiencies of IMF orientation and season depending on the magnetic local time (MLT) location of the stations.

**Comment 18**: Line 76: Rewrite "Donner and co" in different form.

**Response to Comment 18**: In the revised manuscript line 79-84 as: Donner et al. (2019) used RQA to obtain the complementary measures that serves as markers of different physical processes underlying quiet and storm time magnetospheric dynamics. It was shown that the RQA approach discriminates the magnetospheric activity in response to solar wind forcing.

**Comment 19**: "In section 3, we unveiled our results and observation in section 4, the discussion and conclusion in section 5." should be rewritten based on the main body of the manuscript.

**Response to Comment 19**: The above comment has corrected in the revised manuscript as suggested.

**Comment 20**: Line 92: 2.0 → 2

**Response to Comment 20**: the above suggestion has been corrected in the revised manuscript

**Comment 21**: Lines 108, 131, & 135: According to the title in Section 2 (Data Acquisition and the Concept of Nonlinear Dynamics), this section could be organized into two subsections by "Data Acquisition" and "Concept of Nonlinear Dynamics".

**Response to Comment 21**: In the revised manuscript we have organized the section into two subsections as suggested.

**Comment 22**: Equation 3 & Line 124: What does "i" mean in "εi", which has not appeared on Line 111?

**Response to comment 22**: The above expression has been corrected in the revised manuscript line 135-137 as: $R_{i,j} = \Theta\left(\varepsilon - \left\|\overline{x}_i - \overline{x}_j\right\|\right) \quad i,j = 1, 2, \ldots, N$

**Comment 23**: Line 124, Line 134, & Line 140: Are the total number of considered states (N) on Line 124, the number of points (N) on Line 134, and the maximum possible line length in the RP (N) on Line 140 identical? Please also provide the values of N.

**Response to comment 23**: The description of $N$ are concept embedded in the algorithm of the recurrence analysis. There is no specific value.

**Comment 24**:  Lines 129-130: It is NOT necessary to mention "Laminarity (LAM)", "Entropy (ENTR)", "Determinism (DET)", and "Trapping Time (TT)" unless these measures are used in this study. Please rephrase this sentence.

**Response to comment 24**: The above comment has been corrected in the revised manuscript 145-150 as: The Recurrence Quantification Analysis (RQA) is deduced to measure the structure and patterns in the RPs. RQA presents the number and duration of recurrences to characterize the polar cap behaviour. The RQA quantities measures used to reveal the recurrence features in the polar cap index comprises of Recurrence Rate (RR) and Average Diagonal Line Length (L).

**Comment 25**:  Equation 5: i (in the parentheses) $\rightarrow$ l

**Response to Comment 25**: The above suggestion has been corrected in the revised manuscript line 157.

**Comment 26**: Line 139: How do the authors obtain the length of a diagonal line (l)?

**Response to Comment 26**:  how the length of the diagonal line (l) is obtained has been included in the revised manuscript line 160-161 as: The length of a diagonal line is calculated based on the number of recurrence points that form a diagonal line in the RP.

**Comment 27: Line 141 #1: 3.0 $\rightarrow$ 3**

**Response to Comment 27:** The above suggestion has been addressed in the revised manuscript

**Comment 28:** Line 141 #2: Please re-arrange the subsections in Section 3 based on the observations of the polar cap indices and the results obtained from the recurrence plot and recurrence quantification analysis techniques.

**Response to Comment 28**: The section 3 of the revised manuscript as been re-arrange according to suggestion of the above comment.

**Comment 29**: Line 148: Detailed descriptions about "some coherent structure in its underlying dynamics" and "these inherit coherent structures" are needed.

**Response to Comment 29**: In the revised manuscript line 168-172 we have re-written the above comment as: At the different categories of geomagnetic storms corresponding to the polar cap index time series, it was observed that the polar cap signals exhibit some coherent structures, i.e. depictation of self-organized patterns and complex fluctuations signatures in the signals of the polar cap index was evident.  The dynamical information of these complex fluctuation signatures in the signal of the polar cap index is revealed through RP and RQA.

**Comment 30**: Line 188 & Line 203: magenta → red (or orange)?

**Response to comment 30**: The above suggestion has been addressed in the revised manuscript

**Comment 31**: Line 189 & Line 204: grey → purple?

**Response to comment 31**: The above suggestion has been addressed in the revised manuscript

**Comment 32:** Line 213: 4.0 → 4

**Response to comment 32:** The above suggestion has been addressed in the revised manuscript

**Comment 33**: Lines 228-229: What does "similarity dynamics" mean?

**Response to comment 33**: We explained the meaning of "similarity dynamics" in the revised manuscript line 266-273 as: Further results from the recurrence quantification analysis reveals that RR values for PCN and PCS increases with storm intensity. For instance, the highest RR values was observed during great storms, particularly for PCS variations. Severe and strong storms show progressively lower RR values compared to great storms, indicating a gradual increase in similarity dynamics as solar wind energy input strengthens. The similarity dynamics refers to the observed recurring patterns emerging by accounting to the number of consecutive points in the phase space trajectory that exhibit similar behaviour. However, RR values decline significantly during moderate and weak storms, reflecting declining of similarity behaviour in the polar cap activities.

**Comment 34:** Lines 278-280: Delete repetition.

**Response to Comment 34**: The above suggestion has been addressed in the revised manuscript

**Comment 35**: Lines 308-312: Delete repetition.

**Response to Comment 35**: The above suggestion has been addressed in the revised manuscript

**Comment 36**: As mentioned in the first and second minor comments, please specify what kind of "polar cap activities", "polar cap variation", and so on the authors describe.

**Response to Comment 36**: The above suggestion has been addressed in the entire revised manuscript

**Comment 37**: Clear explanations about "deterministic structure" and "deterministic behavior" mentioned in this manuscript are also needed.

**Response to Comment 37**: We have explained the term "deterministic structure" and "deterministic behavior" in the revised manuscript line 212-224 as: Comparatively, these observed wider, wide and high clusters of recurrence points in the RPs for PCN and PCS variation during great, severe, and strong geomagnetic storms signifies that the polar cap exhibits deterministic behaviour as the solar wind energy injection into the magnetosphere intensifies, while the low cluster of recurrence points observed in the RPs during moderate and weak geomagnetic storms reveals that the polar cap exhibits stochastic behaviour. This is attributed to the fact that the

estimation of this recurrence points in the RP is based on recurrence matrix. The matrix compares the states of system at times $i$ and $j$. If the states are similar, the recurrence matrix is indicated by one, ($R_{ij} = 1$), then a black dot is accounted in the RP whenever $R_{ij} = 1$. On the other hand, if the states are different, the corresponding entry in the matrix is zero ($R_{ij} = 0$) with indication of no dot in the RP (Marwan et al. 2007). Therefore, the observed RP for PCN and PCS, where there is intense depiction of black dots in the RP indicates a strong deterministic structure and where there is minima depictation of black dots in the RP indicates a rare deterministic structure. Furthermore, the clusters of recurrence points (i.e. display of many black dots in the RP) observed in the RP for PCN and PCS depicting similarity dynamics in the polar cap variation unveils deterministic behaviour while the low or no cluster of recurrence points in the RPs observed during moderate and weak geomagnetic storm depicting dissimilarities in the polar cap variation unveils stochastic behavior.

**Comment 38**: Quanaaq → Qaanaaq

**Response to Comment 38**: The above suggestion has been addressed in the revised manuscript

**Comment 39**: The words "(EKL)", "(PCN and PCS)", "(PCN)", "(PCS)", "(RPs)", "(RQA)", "(L)", and so on appear many times. Please find and correct the corresponding parts.

**Response to Comment 39**: The above suggestion has been addressed in the revised manuscript

**Comment 40**: The average line length of the diagonal (L) and length of the diagonal (l) could be distinguishable.

**Response to Comment 40**: The above suggestion has been addressed in the revised manuscript

---

## Author Comment (AC2)

**Dear Reviewer**

Thank you for your thorough review of our manuscript. Your comments have helped us to improve the manuscript. Below we provide a point-by-point response to your comments.

**Comment 1:** The authors claim that recurrence analysis has not been applied to polar cap indices (PCN/PCS) in previous literature. In my opinion, this should be demonstrated more explicitly, for example: a short discussion stating what has been done with PC indices, what has been done with RQA in geomagnetism, and where the gap lies.

**Response to Comment 1**: In the revised manuscript (lines 55–100), we have added text to address this suggestion.

For instance, lines 58 of the revised manuscript now read: Several studies have investigated hemispheric differences in the Earth's polar caps using PC indices.

In addition, in lines 78–102 of the revised manuscript we now state that "Additionally, the application of recurrence analysis to geomagnetic activity during geomagnetic storms has been extensively investigated. Oludehinwa et al. (2018) used recurrence analysis to examine nonlinear effects in the disturbance storm time (Dst) index for various categories of geomagnetic storms. They reported that the Dst signals behave in a stochastic manner during minor geomagnetic storms, whereas more deterministic behavior is exhibited during periods of stronger geomagnetic activity. Donner et al. (2018) applied recurrence analysis to reveal the nonlinear features of the hourly Dst index during intervals with magnetic storms and normal variability. They found that recurrence quantification analysis (RQA) distinguishes between storm and non-storm periods even better than other nonlinear characteristics such as symbolic-dynamics-based entropy. Donner et al. (2019) further used RQA to obtain complementary measures that serve as markers of different physical processes underlying quiet- and storm-time magnetospheric dynamics, demonstrating that RQA can discriminate magnetospheric activity in response to solar-wind forcing. Oludehinwa et al. (2021) applied RP and RQA to investigate nonlinear interdependence among solar-wind parameters influencing geomagnetic activity during geomagnetic storms. Solar-wind parameters, including proton density, solar-wind dynamic pressure, IMF Bz, and geomagnetic indices AE and SYM-H, were considered during the pre-storm, storm, and post-storm phases of intense, major, moderate, and minor geomagnetic storms. They found that the RP of the solar-wind parameters display a strong deterministic structure during storms, indicating strong interdependence, whereas during pre-storm and post-storm periods the RP exhibit only rare deterministic structure, signifying weak interdependence.

However, repetitive patterns in polar cap signals in response to varying geomagnetic-storm intensities have not yet been systematically investigated. In particular, to the best of our knowledge, the concept of nonlinear dynamics has not been applied to examine recurrent patterns in the polar cap indices PCN and PCS. Therefore, this study aims to identify recurring patterns in the polar cap north (PCN) and south (PCS) indices in response to different geomagnetic-storm

intensities. We implement recurrence analysis, a nonlinear-dynamics technique that quantifies the times at which a system revisits similar states in its phase-space trajectory. Recurrence plots (RPs) reveal how a complex system returns to similar states over time, whereas recurrence quantification analysis (RQA) provides a set of quantitative measures derived from the RPs to characterize the underlying system dynamics

Comment 2: The present form of the manuscript reads more as a data-driven analysis with limited physical interpretation. The authors do not clearly state why recurrence analysis is the appropriate tool, nor what physical processes they expect to diagnose. In other words, what specific physical mechanisms might produce recurrence in PC indices?

**Response to Comment 2**: In the revised manuscript (lines 42–55), we have added text explaining the physical mechanisms that produce recurrence in the PC indices as follows:

The recurrence observed in the polar cap indices arises from recurring solar-wind input that enhances solar-wind–magnetosphere coupling through magnetic reconnection at the dayside magnetopause. In particular, southward interplanetary magnetic field (IMF Bz) conditions facilitate reconnection, open geomagnetic field lines, and drive enhanced ionospheric convection in the polar caps. These processes increase energy transfer to the magnetosphere and lead to repetitive geomagnetic disturbances that are reflected in elevated PC index values. The resulting rapid variations in the polar cap signals contain coherent structures that are complex, multiscale, and chaotic. These coherent structures are self-organized patterns that emerge from nonlinear interactions between the solar wind, ionosphere, and magnetosphere. They appear in the polar cap index time series as complex fluctuating signatures that exhibit recurrent patterns during different categories of geomagnetic storms, which motivates our analysis. Consequently, complex-systems methods such as recurrence plots (RPs) and recurrence quantification analysis (RQA), which characterize nonlinear processes in dynamical systems, are well suited to capture these recurring patterns in the polar cap signals.

Comment 3: The authors state that the recurrence analysis was computed using the embedding dimension (m=15) and time delay (\tau=6). Without justification, the results may not be meaningful. It should be clarified why/how such values have been chosen, especially the utilized such a high embedding dimension. Did the authors employ a standard method (e.g., False Nearest Neighbors for m or Average Mutual Information for \tau) to justify these specific values? Justifying these core parameters is essential for the reproducibility and validity of the nonlinear analysis.

**Response to Comment 3**: In the revised manuscript lines 156-163, we included statement that explicitly explain how the recurrence analysis was computed and also show the figures of false nearest neighbour and average mutual information as:

"Our recurrence analysis was computed using a threshold distance  $\varepsilon = 1.5$ , embedding dimension m = 15, and time delay  $\tau = 6$ , to reveal the polar cap recurrence pattern during different categories of geomagnetic storms shown in Table 1. The embedding dimension m and time delay

 $\tau$  were determined using the false nearest neighbors and average mutual information methods, respectively, as shown in Figures 2 and 3. The optimal embedding dimension was chosen where the fraction of false nearest neighbors reaches a stable low value as m increases, whereas the time delay was selected at the first local minimum of the average mutual information curve. Recurrence quantification analysis (RQA) was then used to measure the structure and patterns in the RPs. RQA provides information on the number and duration of recurrences in order to characterize the behavior of the polar cap indices. In this study, we focus on two RQA measures to describe the recurrence features of the PC indices: the recurrence rate (RR) and the average diagonal line length (L)."

Figure 2: The plot of Average Mutual Information for the PCN and PCS index

Figure 3: The plot of False Nearest Neighbors for the PCN and PCS index

**Comment 4**: A significant clarification is required regarding other parameters used in the recurrence analysis. For example, threshold (\epsilon) choice strongly influences RP density, RR, L, and all subsequent claims. Additionally, how robust are the obtained results to changes in these parameters (m, \tau, \epsilon, N)?

Response to Comment 4: We have included the value of the threshold  $\varepsilon$  in the revised manuscript. Specifically, our recurrence analysis was computed using a threshold distance  $\varepsilon = 1.5$ , embedding dimension m = 15, and time delay  $\tau = 6$  to reveal the polar cap recurrence patterns for the different categories of geomagnetic storms, as summarized in Table 1.

**Comment 5**: PCN/PCS amplitude increases during stronger storms, which automatically increases recurrence density unless the time series is normalized. Thus, the manuscript risks conflating amplitude effects with dynamical structure. I would recommend to normalize the PCN/PCS time series to avoid amplitude-driven RR inflation.

**Response to Comment 5**: In the revised manuscript, we have normalized the PCN/PCS time series and then recalculated the recurrence analysis of the PC indices. We found that the recurrence analysis results remain essentially the same. The PCN/PCS time series was normalized using the expression given below and has been included in line 135 of the revised manuscript

$$x_i = \frac{x - x_{mean}}{x_{std}}$$

Below we show bar charts of the recurrence-analysis results for the normalized PCN/PCS time series:

Figure A: Bar chart of recurrence rate (RR) for the normalized PCN and PCS time series during great, severe, strong, moderate, and weak geomagnetic storms

Figure B: Bar chart of average diagonal line length (L) for the normalized PCN and PCS time series during great, severe, strong, moderate, and weak geomagnetic storms

For comparison, we also show bar charts of the recurrence-analysis results for the original (unnormalized) PCN/PCS time series:

Figure 9: The Bar Chart of Length of Diagonal Line (L) for PCN and PCS at great storm, severe storm, strong storm, moderate storm and weak geomagnetic storm.

Figure 10: The Bar Chart of Recurrence Rate (RR) for PCN and PCS at great storm, severe storm, strong storm, moderate storm and weak geomagnetic storm.

This confirms that the differences in RR and L across the different categories of geomagnetic storms are not dominated by amplitude effects but instead reflect differences in the underlying dynamical structure

**Comment 6**: L110:  $(i,l) \rightarrow (i,j)$

**Response to Comment 6**: In the revised manuscript (line 140), we have corrected the expression as: The RPs is based on recurrence matrix where each element (i, j)

Comment 7: L131: "It is mathematically express as" ---> "... expressed as"

**Response to Comment 7**: In the revised manuscript (line 140), we have corrected the sentence to: 'It is mathematically expressed as:'

Comment 8: L214: "The RP results... reveals distinct patterns" ---> "... reveal ..."

**Response to Comment 8**: In the revised manuscript line 269, we have corrected the comment as: The RP results of the polar cap activities during geomagnetic storms **reveal** distinct patterns tied to storm intensity.

Comment 9: L214: "Great storms, severe storms, and strong storms reveals..." ---> "... reveal ..."

**Response to Comment 9**: In the revised manuscript lines 269-270, we have corrected the comment as: RP of PCN and PCS corresponding to great storms, severe storms, and strong storms **reveal** a robust deterministic structure in RPs.

Comment 10: L234: "...categories of suggests that..." The noun is missing after "categories of"

**Response to Comment 10**: In the revised manuscript lines 291-293, we have corrected the comment as: This observation of L values in the RPs during different categories of **geomagnetic storms** suggests that the polar cap activities are highly chaotic during moderate and weak storms compared to strong, severe and great storms.

**Comment 11**: L138 (Eq.5): Shouldn't be P(l) not P(i)?

**Response to Comment 11**: Thank you for the observation. The equation has been cross-checked; the expression is correct.

**Comment 12:** Several references are duplicated (e.g., Oludehinwa 2018, Donner 2019).

**Response to Comment 12:** The references have been cross-checked. In some cases (e.g., Oludehinwa et al., 2018; Donner et al., 2019), the same authors have multiple publications, but these entries correspond to different papers with distinct years and titles.